

# The ethnobotany of psychoactive plant use: a phylogenetic perspective

Nashmiah Aid Alrashedy[*] and Jeanmaire Molina[*]

Department of Biology, Long Island University, Brooklyn, NY, United States
[*] These authors contributed equally to this work.

## ABSTRACT

Psychoactive plants contain chemicals that presumably evolved as allelochemicals but target certain neuronal receptors when consumed by humans, altering perception, emotion and cognition. These plants have been used since ancient times as medicines and in the context of religious rituals for their various psychoactive effects (e.g., as hallucinogens, stimulants, sedatives). The ubiquity of psychoactive plants in various cultures motivates investigation of the commonalities among these plants, in which a phylogenetic framework may be insightful. A phylogeny of culturally diverse psychoactive plant taxa was constructed with their psychotropic effects and affected neurotransmitter systems mapped on the phylogeny. The phylogenetic distribution shows multiple evolutionary origins of psychoactive families. The plant families Myristicaceae (e.g., nutmeg), Papaveraceae (opium poppy), Cactaceae (peyote), Convolvulaceae (morning glory), Solanaceae (tobacco), Lamiaceae (mints), Apocynaceae (dogbane) have a disproportionate number of psychoactive genera with various indigenous groups using geographically disparate members of these plant families for the same psychoactive effect, an example of cultural convergence. Pharmacological traits related to hallucinogenic and sedative potential are phylogenetically conserved within families. Unrelated families that exert similar psychoactive effects also modulate similar neurotransmitter systems (i.e., mechanistic convergence). However, pharmacological mechanisms for stimulant effects were varied even within families suggesting that stimulant chemicals may be more evolutionarily labile than those associated with hallucinogenic and sedative effects. Chemically similar psychoactive chemicals may also exist in phylogenetically unrelated lineages, suggesting convergent evolution or differential gene regulation of a common metabolic pathway. Our study has shown that phylogenetic analysis of traditionally used psychoactive plants suggests multiple ethnobotanical origins and widespread human dependence on these plants, motivating pharmacological investigation into their potential as modern therapeutics for various neurological disorders.

Corresponding author
Jeanmaire Molina,
jeanmaire.molina@liu.edu

## INTRODUCTION

Plants constantly evolve to produce various defensive secondary metabolites against their equally adaptive predators (*Polya, 2003*; *Wink, 2003*; *Wink, 2016*). Some well-known psychoactive compounds such as atropine, caffeine, cocaine, nicotine and

morphine are believed to have been products of this evolutionary arms race (*Howe & Jander, 2008*; *Fürstenberg-Hägg, Zagrobelny & Bak, 2013*). Psychoactive, alternatively psychotropic, substances act on the nervous system affecting mental processes and behavior (*Spinella, 2001*; *Rätsch, 2005*). They include hallucinogens that distort reality, sedatives/narcotics that induce sleep, calmative or anxiolytics, antidepressants, and stimulants that wake the mind (*Spinella, 2001*; *Rätsch, 2005*; *Van Wyk & Wink, 2014*). Interestingly, humans have exploited alternate uses for plants containing psychoactive phytochemicals that have purportedly evolved to ward off plant predators. However, the affinity of these phytochemicals within the hominid nervous system may also indicate some kind of mutualistic co-evolution, with ancient humans seeking and perhaps cultivating plant psychotropics to facilitate survival, by alleviating starvation, fatigue and pain (*Sullivan & Hagen, 2002*).

Psychoactive compounds have specific molecular targets in the nervous system, and interact in a particular way with neuronal receptors to produce various psychoactive effects (*Spinella, 2001*; *Polya, 2003*). For instance, morphine in opium poppy (*Papaver somniferum*, Papaveraceae) eliminates pain by binding to opioid receptors (*Polya, 2003*), but simultaneously promotes sedation and euphoria, by disinhibiting dopamine-containing neurons in the limbic system (*Johnson & North, 1992*). Dopamine is ultimately responsible for feelings of elation and satisfaction, which occur after some rewarding act like sex or food satiety. Addiction arises from wanting to re-experience the pleasure due to the drug's ability to cause dopamine buildup (*Lüscher & Ungless, 2006*). Compounds that mimic serotonin and act as receptor agonists like mescaline in the peyote cactus (*Lophophora williamsii*, Cactaceae), trigger hallucinations and cognitive breakdown (*Polya, 2003*). Stimulating substances, such as the alkaloid nicotine in tobacco, *Nicotiana tabacum* (Solanaceae), mimic the endogenous neurotransmitter acetylcholine stimulating muscle contractions and cholinergic areas of the brain involved in arousal and attention (*Polya, 2003*). Yet, the confamilial *Atropa belladonna*, contains a chemically different alkaloid, atropine, which promotes sedation and incapacitation via its action as muscarinic acetylcholine antagonist, blocking neuromuscular communication (*Spinella, 2001*).

It is well established that all cultures, ancient or modern, have some kind of drug culture, relying on psychoactives for recreational, ritual and/or medicinal uses (*Schultes, 1976*; *Schultes, Hofmann & Rätsch, 2001*; *Rätsch, 2005*). Shamanistic religions have existed in the Old World of Europe, Asia and Africa, believing that psychoactive plants are capable of healing through divine power. Marijuana (*Cannabis* spp., Cannabaceae) and opium poppy are among the most popular psychoactive plants used by Old World shamans. Marijuana was used in ancient China for various afflictions like malaria and constipation, and even as a narcotic in surgeries. In India, the plant was considered sacred promoting pleasurable sensations in the user (*Clarke & Merlin, 2013*). Tetrahydrocannabinol (THC) in marijuana, exerts these actions by binding to cannabinoid receptors, mediating sensory pleasure (*Mahler, Smith & Berridge, 2007*). Another familiar psychoactive, opium poppy was used for medicinal and recreational purposes. It probably originated in the Mediterranean, but widespread use has confounded its evolutionary origin (*Merlin, 2003*). It was recorded in the Eber papyrus, an ancient Egyptian scroll, that opium poppy was used to stop the

excessive crying of children (*Vetulani, 2001*). The plant contains morphine and codeine that are responsible for its hypnotic and analgesic properties (*Heinrich et al., 2012*).

Indigenous people of the New World have also used psychotropic substances, including tobacco, ayahuasca, and coca, even more so than cultures of the Old World (*Schultes, 1976*). Tobacco from the leaves of *N. tabacum* has long been used in the Americas, with cultivation in pre-Columbian Mexico or Peru (*Rätsch, 2005*). American Indians believed in the medicinal power of tobacco, and it was smoked in ceremonial peace pipes to seal covenants. In the Amazon Basin of South America, the hallucinogenic beverage, ayahuasca, is made by healers from the boiled crushed stems of the caapi, *Banisteropsis caapi* (Malpighiaceae), along with the leaves of chacruna, *Psychotria viridis* (Rubiaceae). Chacruna contains serotonergic N, N-dimethyltryptamine (DMT), that is activated by the beta-carbolines in caapi (*McKenna, 1996*). In the Andes, indigenous peoples chew coca leaves of *Erythroxylum coca* (Erythroxylaceae) to cope with hard labor, removing symptoms of fatigue and hunger (*Nigg & Seigler , 1992*). Its cocaine content prevents dopamine reuptake producing increased energy and mood elevation (*Spinella, 2001*).

The ubiquity of psychoactive plants in various cultures motivates investigation of the commonalities among these plants, in which a phylogenetic framework may be insightful. Information is assigned to nodes of the phylogeny, instead of one species at a time, facilitating the study of trait distributions (*Saslis-Lagoudakis et al., 2015*). Phylogenetic studies of culturally diverse medicinal plants have repeatedly shown that medicinal uses and phytochemical traits are not randomly distributed on the phylogeny, but are shared by closely related plants, regardless of these plants' cultural and geographic designations (*Saslis-Lagoudakis et al., 2012*; *Saslis-Lagoudakis et al., 2015*; *Xavier & Molina, 2016*). In this study we aimed to understand if there is a similar pattern of cultural convergence (*Xavier & Molina, 2016*) in psychoactive plants using phylogenetic analysis—does the phylogeny of culturally important psychoactive plants reveal a preference for certain plant families and for specific psychoactive effects (hallucinogenic, sedative, stimulant, etc.)? Additionally, we sought to understand if there is also a pattern of mechanistic convergence, such that unrelated plants with similar psychoactive effects ultimately affect similar neurotransmitter systems. Our study provides insight into the ethnobotanical origins of psychoactive plant use and suggests new plant sources of psychopharmacological drugs.

## MATERIALS AND METHODS

Pyschoactive taxa of seed plants (126 genera) used by various indigenous groups were compiled for this study (Table 1), but plants with psychoactive uses only after alcoholic fermentation were excluded (e.g., wine from grapes, *Vitis vinifera*). Congeneric species were only represented once in the phylogeny, e.g., *Datura* spp. included *D. discolor* Bernh., *D. ferox* L., *D. innoxia* Mill., *D. metel* L., *D. stramonium* L., *D. wrightii* Regel. This is to account for taxonomic uncertainties that are common in species circumscriptions, and also not to visually bias the phylogeny towards a certain family with multi-species genera (e.g., *Datura)*. The plants' names were verified in (*The Plant List, 2013*), a working list

Alrashedy and Molina (2016), *PeerJ*, DOI 10.7717/peerj.2546

**Table 1  Psychoactive plant taxa in this study.** Culturally diverse psychoactive plant taxa, their uses, indigenous psychoactive cultural origin, and corresponding Genbank numbers.

| Family (Order) | Accepted binomial name | Common name | Indigenous psychoactive culture | Mechanism of action | Genbank numbers |
|---|---|---|---|---|---|
| Acanthaceae (Lamiales) | *Justicia pectoralis* Jacq. | justicia | Native American (*Rätsch, 2005*) | Hallucinogen, antidepressant, sedative, aphrodisiac (*Rätsch, 2005*) | AJ879453 |
| Acoraceae (Acorales) | *Acorus calamus* L. | sweet flag | Indomalayan, Temperate Asian (*Rätsch, 2005*) | Stimulant, antidepressant, sedative (*Rätsch, 2005*) | AJ879453 |
| Aizoaceae (Caryophyllales) | *Sceletium* spp. | kougoed | African and Middle Eastern (*Gericke & Viljoen, 2008*) | Sedative, analgesic (*Gericke & Viljoen, 2008*) | HM850175 |
| Apiaceae (Apiales) | *Angelica sinensis* (Oliv.) Diels | dong quai | Temperate Asian (*Rätsch, 2005*) | Stimulant, sedative (*O'Mahony, 2010*) | GQ436632 |
| Apiaceae (Apiales) | *Centella asiatica* (L) Urb. | gotu kola | Indomalayan, Temperate Asian (*Rätsch, 2005*) | Antianxiety, antidepressant (*Mamedov, 2005*) | GQ436635 |
| Apocynaceae (Gentianales) | *Alstonia scholaris* (L.) R. Br. | dita | African and Middle Eastern, Australasian, Indomalayan (*Rätsch, 2005*; *Arulmozhi et al., 2012*) | Stimulant, antianxiety, antidepressant, sedative, analgesic, aphrodisiac (*Rätsch, 2005*; *Arulmozhi et al., 2012*) | EU916739 |
| Apocynaceae (Gentianales) | *Apocynum venetum* L | luobuma | Temperate Asian (*Grundmann et al., 2007*) | Antianxiety, antidepressant (*Grundmann et al., 2007*; *Zheng, Fan & Liu, 2013*) | KP088474 |
| Apocynaceae (Gentianales) | *Carissa edulis* (Forssk.) Vahl | Arabian numnum | African and Middle Eastern (*Rätsch, 2005*) | Hallucinogen, aphrodisiac (*Rätsch, 2005*) | JF265327 |
| Apocynaceae (Gentianales) | *Rauvolfia serpentina* (L.) Benth. ex Kurz | snakeroot | Indomalayan (*Mamedov, 2005*) | Antianxiety, antidepressant, sedative, analgesic (*Spinella, 2001*; *Mamedov, 2005*; *Rätsch, 2005*) | KJ667614 |
| Apocynaceae (Gentianales) | *Tabernaemontana* spp. | milkweed | Indomalayan, African, Native American (*Rätsch, 2005*) | Hallucinogen, stimulant, antidepressant, sedative, analgesic (*Rätsch, 2005*; *Pratchayasakul et al., 2008*; *Cardoso et al., 2015*) | EU916740 |
| Apocynaceae (Gentianales) | *Tabernanthe iboga* Baill. | iboga | African and Middle Eastern (*Sayin, 2014*) | Hallucinogen, stimulant, antianxiety, antidepressant, analgesic, aphrodisiac (*Nigg & Seigler, 1992*; *Sayin, 2014*) | AJ419763 |

Peer J

**Table 1** (*continued*)

| Family (Order) | Accepted binomial name | Common name | Indigenous psychoactive culture | Mechanism of action | Genbank numbers |
|---|---|---|---|---|---|
| Apocynaceae (Gentianales) | *Voacanga* spp. | voacango bush | African and Middle Eastern (*Rätsch, 2005*) | Hallucinogen, stimulant, aphrodisiac (*Rätsch, 2005*) | KC628529 |
| Aquifoliaceae (Aquifoliales) | *Ilex* spp. | yerba mate | Native American (*Rätsch, 2005*) | Stimulant (*Rätsch, 2005*) | FJ394625 |
| Araliaceae (Apiales) | *Panax ginseng* C.A.Mey. | ginseng | Temperate Asian (*Rätsch, 2005*) | Stimulant, antidepressant, aphrodisiac (*Rätsch, 2005*) | KM088019 |
| Arecaceae (Arecales) | *Areca catechu* L. | betel nut | Indomalayan (*Rätsch, 2005*) | Stimulant, sedative, aphrodisiac (*Rätsch, 2005*) | JX571781 |
| Asteraceae (Asterales) | *Artemisia* spp. | wormwood | European; Temperate Asian (*Rätsch, 2005*; *Sayin, 2014*) | Hallucinogen, stimulant, analgesic aphrodisiac (*Rätsch, 2005*; *Sayin, 2014*) | KM360653 |
| Asteraceae (Asterales) | *Calea ternifolia* Oliv | dream herb | Native American (*Rätsch, 2005*) | Hallucinogen, sedative (*Rätsch, 2005*) | AY215089 |
| Asteraceae (Asterales) | *Lactuca virosa* Habl. | wild lettuce | African and Middle Eastern (*Rätsch, 2005*) | Sedative, aphrodisiac (*Rätsch, 2005*) | KM360888 |
| Asteraceae (Asterales) | *Tagetes* spp. | Mexican marigold | Native American (*Rätsch, 2005*) | Hallucinogen, stimulant, antianxiety, antidepressant, aphrodisiac (*Rätsch, 2005*) | AY215184 |
| Bignoniaceae (Lamiales) | *Bignonia nocturna* (Barb.Rodr.) L.G.Lohmann [=*Tanaecium nocturnum* (Barb.Rodr.) Burea & K.Schum.] | koribo | Native American (*Rätsch, 2005*) | Sedative, analgesic and aphrodisiac (*Rätsch, 2005*) | KR534325 |
| Burseraceae (Sapindales) | *Boswellia sacra* Flueck. | olibanum tree | African and Middle Eastern (*Rätsch, 2005*) | Hallucinogen (*Rätsch, 2005*) | KT934315 |
| Cactaceae (Caryophyllales) | *Ariocarpus fissuratus* (Engelm.) K.Schum. | chautle | Native American (*Rätsch, 2005*) | Hallucinoge, analgesic (*Rätsch, 2005*) | KC777009 |
| Cactaceae (Caryophyllales) | *Echinopsis* spp. (incl. *Trichocereus pachanoi* Britton & Rose) | San Pedro cactus | Native American (*Rätsch, 2005*) | Hallucinogen, stimulant (*Rätsch, 2005*) | FR853367 |

Alrashedy and Molina (2016), *PeerJ*, DOI 10.7717/peerj.2546

**Table 1** (*continued*)

| Family (Order) | Accepted binomial name | Common name | Indigenous psychoactive culture | Mechanism of action | Genbank numbers |
|---|---|---|---|---|---|
| Cactaceae (Caryophyllales) | *Lophophora williamsii* (Lem. Ex Salm-Dyck) J.M. Coult. | peyote | Native American (*Vetulani, 2001*) | Hallucinogen (*Vetulani, 2001*) | KC777011 |
| Cactaceae (Caryophyllales) | *Mammillaria* spp. | false peyote | Native America (*Rätsch, 2005*) | Hallucinogen (*Rätsch, 2005*) | KC777008 |
| Cactaceae (Caryophyllales) | *Pachycereus pecten-aboriginum* (Engelm. ex S. Watson) Britton & Rose | pitayo | Native American (*Schultes, 1976*) | Hallucinogen (*Schultes, 1976*) | JN191499 |
| Campanulaceae (Asterales) | *Lobelia tupa* L. | tupa | Native American (*Schultes, 1976*) | Hallucinogen, sedative (*Schultes, 1976*; *Rätsch, 2005*) | EF174606 |
| Cannabaceae (Rosales) | *Cannabis* spp. | marijuana | Indomalayan, Temperate Asian (*Rätsch, 2005*) | Hallucinogen, stimulant, antianxiety, antidepressant, sedative, analgesic, aphrodisiac (*Rätsch, 2005*) | AF500344 |
| Cannabaceae (Rosales) | *Humulus lupulus* L. | hops | European (*Rätsch, 2005*) | Antianxiety, sedative (*Heinrich et al., 2012*) | KT266264 |
| Caprifoliaceae (Dipsacales) | *Nardostachys jatamansi* (D. Don) DC. | jatamansi | Indomalaya (*Chaudhary et al., 2015*) | Antidepressant, sedative (*Chaudhary et al., 2015*) | AF446950 |
| Caprifoliaceae (Dipsacales) | *Valeriana officinalis* L. | valerian | European (*Heinrich et al., 2012*) | Antianxiety and sedative (*Heinrich et al., 2012*) | AY362490 |
| Celastraceae (Calastrales) | *Catha edulis* (Vahl) Endl. | khat | African and Middle Eastern (*Rätsch, 2005*) | Stimulant, antidepressant, aphrodisiac (*Rätsch, 2005*) | JQ412336 |
| Columelliaceae (Bruniales) | *Desfontainia spinosa* Ruiz & Pav. | taique | Native American (*Rätsch, 2005*) | Hallucinogen (*Rätsch, 2005*) | Z29670 |
| Combretaceae (Myrtales) | *Terminalia bellirica* (Gaertn.) Roxb. | bellerian myrobalan | Indomalaya (*Rätsch, 2005*) | Hallucinogen, sedative (*Rätsch, 2005*) | KT279740 |
| Convolvulaceae (Solanales) | *Argyreia nervosa* (Burm. F.) Bojer (=*Argyreia speciosa* (L. f.) Sweet) | Hawaiian baby | Native American (*Rätsch, 2005*) | Hallucinogen, analgesic, aphrodisiac (*Rätsch, 2005*; *Galani, Patel & Patel, 2010*) | KF242477 |
| Convolvulaceae (Solanales) | *Convolvulus tricolor* L. | dwart morning glory | European (*Rätsch, 2005*) | Sedative, analgesic (*Rätsch, 2005*) | L11683 |

**Table 1** (*continued*)

| Family (Order) | Accepted binomial name | Common name | Indigenous psychoactive culture | Mechanism of action | Genbank numbers |
|---|---|---|---|---|---|
| Convolvulaceae (Solanales) | *Ipomoea* spp. | morning glory | Native American (*Rätsch, 2005*) | Hallucinogen, stimulant, aphrodisiac (*Rätsch, 2005*; *Meira et al., 2012*) | KF242478 |
| Convolvulaceae (Solanales) | *Turbina corymbosa* (L.) Raf. | ololiuqui vine | Native American (*Rätsch, 2005*) | Hallucinogen, analgesic (*Rätsch, 2005*) | AY100966 |
| Cupressaceae (Pinales) | *Juniperus recurva* Buch.-Ham. ex D. Don | Himalayan weeping juniper | Indomalayan, Temperate Asian (*Rätsch, 2005*) | Hallucinogen (*Rätsch, 2005*) | JQ512552 |
| Ephedraceae (Ephedrales) | *Ephedra* spp. | ephedra | Temperate Asian (*Heinrich et al., 2012*) | Stimulant (*Rätsch, 2005*) | AY056562 |
| Ericaceae (Ericales) | *Ledum palustre* L. | wild rose-mary | Temperate Asian (*Rätsch, 2005*) | Hallucinogen, sedative, analgesic (*Rätsch, 2005*) | AF419831 |
| Ericaceae (Ericales) | *Rhododendron moll*e G.Don. | yang zhi zhu | Temperate Asian (*Mamedov, 2005*) | Antidepressant (*Mamedov, 2005*) | AF421101 |
| Erythroxylaceae (Malpighiales) | *Erythroxylum* spp. | Coca | Native American (*Rätsch, 2005*) | Stimulant, antianxiety, analgesic and aphrodisiac (*Rätsch, 2005*) | AB925614 |
| Fabaceae (Fabales) | *Acacia* spp. | wattle | African/Middle Eastern Australasian, Indomalayan, Native American (*Rätsch, 2005*) | Hallucinogen, aphrodisiac (*Rätsch, 2005*) | HM849736 |
| Fabaceae (Fabales) | *Anadenanthera* spp. | vilca, yopo | Native American (*Rätsch, 2005*) | Hallucinogen and analgesic (*Schultes, 1976*) | KJ082119 |
| Fabaceae (Fabales) | *Astragalus* spp. | milk vetch | Native America (*Rätsch, 2005*) | Hallucinogen (*Rätsch, 2005*) | KU666554 |
| Fabaceae (Fabales) | *Calliandra anomala* (Kunth) J.F. Macbr. | cabellito | Native American (*Rätsch, 2005*) | Hallucinogen and analgesic (*Rätsch, 2005*) | AM234255 |
| Fabaceae (Fabales) | *Desmanthus illinoensis* (Michx.) MacMill. | prairie bundle flower | Native American (*Halpern, 2004*) | Hallucinogen (*Halpern, 2004*) | KP126868 |
| Fabaceae (Fabales) | *Erythrina* spp. | coral trees | Native American, Indomalaya (*Rätsch, 2005*). | Hallucinogen and sedative (*Rätsch, 2005*) | AB045801 |
| Fabaceae (Fabales) | *Lonchocarpus violaceus* Benth. | balche' tree | Native American (*Rätsch, 2005*) | Hallucinogen (*Rätsch, 2005*) | JQ626245 |
| Fabaceae (Fabales) | *Mimosa* spp. | mimosa | Native American, Indomalayan (*Rätsch, 2005*) | Hallucinogenic, sedative, aphrodisiac (*Rätsch, 2005*) | KJ773686 |

Peerj

Table 1 (*continued*)

| Family (Order) | Accepted binomial name | Common name | Indigenous psychoactive culture | Mechanism of action | Genbank numbers |
|---|---|---|---|---|---|
| Fabaceae (Fabales) | *Mucuna pruriens* (L.) DC. | velvet bean | Indomalayan (*Lampariello et al., 2012*) | Hallucinogen, aphrodisiac (*O'Mahony, 2010*; *Lampariello et al., 2012*) | EU128734 |
| Fabaceae (Fabales) | *Rhynchosia pyramidalis* (Lam.) Urb. | bird's eyes | Native American (*Rätsch, 2005*) | Sedative (*Rätsch, 2005*) | KJ594450 |
| Fabaceae (Fabales) | *Sophora secundiflora* (Ortega) DC. | mescal bean | Native American (*Schultes, 1976*) | Hallucinogen (*Schultes, 1976*) | Z70141 |
| Hypericaceae (Malpighiales) | *Hypericum perforatum* L. | St. John's wort | European (*Spinella, 2001*) | Antianxiety, antidepressant (*Spinella, 2001*; *Heinrich et al., 2012*) | AF206779 |
| Iridaceae (Asparagales) | *Crocus sativus* L. | saffron | European (*Rätsch, 2005*) | Antianxiety, sedative, aphrodisiac (*Rätsch, 2005*; *Hossein-zadeh & Noraei, 2009*) | KF886671 |
| Lamiaceae (Lamiales) | *Lavandula angustifolia* Mill. (=*Lavandula officinalis* Chaix) | lavender | European (*Rätsch, 2005*) | Antianxiety, sedative, analgesic (*Lis-Balchin & Hart, 1999*; *Hajhashemi, Ghannadi & Sharif, 2003*) | KT948988 |
| Lamiaceae (Lamiales) | *Leonotis leonurus* (L.) R. Br. | lion's tail | African and Middle Eastern (*Rätsch, 2005*) | Hallucinogen, sedative, analgesic (*Rätsch, 2005*) | AM234998 |
| Lamiaceae (Lamiales) | *Leonurus cardiaca* L. | motherwort | European (*Rauwald et al., 2015*) | Antianxiety, antidepressant, sedative (*Rauwald et al., 2015*) | KM360848 |
| Lamiaceae (Lamiales) | *Melissa officinalis* L. | lemon balm | European (*Vogl et al., 2013*) | Antianxiety, sedative (*Heinrich et al., 2012*) | KM360879 |
| Lamiaceae (Lamiales) | *Plectranthus scutellarioides* (L.) R.Br. (=*Coleus blumei* Benth.) | coleus | Indomalayan (*Rätsch, 2005*) | Hallucinogen, analgesic (*Rätsch, 2005*) | JQ933273 |
| Lamiaceae (Lamiales) | *Rosmarinus officinalis* L. | rosemary | European (*Ferlemi et al., 2015*) | Antianxiety, antidepressant, analgesic (*Ferlemi et al., 2015*) | KR232566 |
| Lamiaceae (Lamiales) | *Salvia divinorum* Epling & Jativa | yerba de la pastora | Native American (*Rätsch, 2005*) | Hallucinogen, analgesic (*Rätsch, 2005*) | AY570410 |
| Lamiaceae (Lamiales) | *Scutellaria lateriflora* L. | skullcap | Native American (*Awad et al., 2003*) | Antianxiety, sedative (*Awad et al., 2003*) | HQ590266 |
| Lauraceae (Laurales) | *Cinnamomum camphora (L.)* J. Presl | camphor | Indomalayan, Temperate Asian (*Rätsch, 2005*) | Stimulant, sedative (*Rätsch, 2005*) | L12641 |

**Table 1** (*continued*)

| Family (Order) | Accepted binomial name | Common name | Indigenous psychoactive culture | Mechanism of action | Genbank numbers |
|---|---|---|---|---|---|
| Lauraceae (Laurales) | *Sassafras albidum* (Nutt.) Nees | sassafras | Native American (*Rätsch, 2005*) | Stimulant (*Rätsch, 2005*) | AF206819 |
| Loganiaceae (Gentianales) | *Strychnos nux-vomica* L. | strychnine tree | Indomalaya (*Rätsch, 2005*) | Stimulant, antianxiety, antidepressant, aphrodisiac (*Rätsch, 2005*) | L14410 |
| Lythraceae (Myrtales) | *Heimia salicifolia* (Kunth) Link | sinicuiche | Native American (*Rätsch, 2005*) | Hallucinogen, sedative (*Rätsch, 2005*) | AY905410 |
| Malpighiaceae (Malpighiales) | *Banisteriopsis* spp. | ayahuasca | Native American (*Sayin, 2014*) | Hallucinogen (*Sayin, 2014*) | HQ247440 |
| Malpighiaceae (Malpighiales) | *Diplopterys cabrerana* (Cuatrec) B. Gates | chaliponga | Native American (*Sayin, 2014*) | Hallucinogen (*O'Mahony, 2010*) | HQ247482 |
| Malvaceae (Malvales) | *Cola* spp. | kola nut | Africa and Middle Eastern (*McClatchey et al., 2009*) | Stimulant (*McClatchey et al., 2009*) | AY082353 |
| Malvaceae (Malvales) | *Sida acuta* Burm.f. | broomweed | Native America (*Rätsch, 2005*) | Stimulant (*Rätsch, 2005*) | KJ773888 |
| Malvaceae (Malvales) | *Theobroma* spp. | cacao | Native American (*Rätsch, 2005*) | Stimulant (*Rätsch, 2005*) | JQ228389 |
| Malvaceae (Malvales) | *Tilia* spp. | linden | European (*Rätsch, 2005*) | Antianxiety, sedative (*Rätsch, 2005*) | KT894775 |
| Melanthiaceae (Liliales) | *Veratrum album* L. | white hellebore | European (*Rätsch, 2005*) | Hallucinogen (*Rätsch, 2005*) | KM242984 |
| Myristicaceae (Magnoliales) | *Horsfieldia australiana* S. T. Blake | nutmeg | Australasian (*Rätsch, 2005*) | Hallucinogen (*Rätsch, 2005*) | KF496315 |
| Myristicaceae (Magnoliales) | *Myristica fragrans* Houtt. | nutmeg | Australiasia, Indomalaya (*Rätsch, 2005*) | Hallucinogen, stimulant, sedative aprhodisiac (*Rätsch, 2005*) | AF206798 |
| Myristicaceae (Magnoliales) | *Osteophloeum platyspermum* (Spruce ex A.DC.) Warb. | huapa | Native American (*Rätsch, 2005*) | Hallucinogen (*Rätsch, 2005*) | JQ625884 |
| Myristicaceae (Magnoliales) | *Virola elongata* (Benth.) Warb. | epena | Native American (*Rätsch, 2005*) | Hallucinogen, stimulant (*Rätsch, 2005*) | JQ626043 |
| Myrtaceae (Myrtales) | *Psidium guajava* L. | guava | African and Middl Eastern (*Rätsch, 2005*) | Sedative, analgesic (*Rätsch, 2005*) | JQ025077 |
| Nitrariaceae (Sapindales) | *Peganum harmala* L. | harmal | African and Middle Eastern (*Sayin, 2014*) | Hallucinogen, stimulant, analgesic (*Vetulani, 2001*; *Farouk et al., 2008*) | DQ267164 |

**Table 1** (*continued*)

| Family (Order) | Accepted binomial name | Common name | Indigenous psychoactive culture | Mechanism of action | Genbank numbers |
|---|---|---|---|---|---|
| Nymphaeaceae (Nymphaeales) | *Nuphar lutea* (L.) Sm. | yellow water lily | European (*Rätsch, 2005*) | Sedative (*Rätsch, 2005*) | DQ182338 |
| Nymphaeaceae (Nymphaeales) | *Nymphaea* spp. | water lily | African and Middle Eastern (*Rätsch, 2005*) | Sedative (*Rätsch, 2005*) | GQ468660 |
| Olacaceae (Santalales) | *Ptychopetalum olacoides* Benth. | marapuama | Native American (*Piato et al., 2008*) | Stimulant, Antidepressant (*Piato et al., 2008*) | FJ038139 |
| Orchidaceae (Asparagales) | *Vanilla planifolia* Jacks. ex Andrews | vanilla | Native America (*Rätsch, 2005*) | Stimulant, sedative, aphrodisiac (*Rätsch, 2005; O'Mahony, 2010*) | KJ566306 |
| Orobanchaceae (Lamiales) | *Cistanche deserticola* K.C.Ma | rou cong rong | Temperate Asian (*Wang, Zhang & Xie, 2012*) | Stimulant, aphrodisiac (*O'Mahony, 2010*) | KC128846 |
| Pandanaceae (Pandanales) | *Pandanus* spp. | screwpine | Australasian (*Rätsch, 2005*) | Hallucinoge, analgesic (*Rätsch, 2005*) | JX903247 |
| Papaveraceae (Ranunculales) | *Argemone mexicana* L. | Mexican poppy | Native American (*Rätsch, 2005*) | Hallucinogen, sedative, analgesic, aphrodisiac (*Rätsch, 2005; Brahmachari, Gorai & Roy, 2013*) | U86621 |
| Papaveraceae (Ranunculales) | *Eschscholzia californica* Cham. | California poppy | Native American (*Rolland et al., 1991*) | Antianxiety, sedative, analgesic (*Rolland et al., 1991*) | KM360775 |
| Papaveraceae (Ranunculales) | *Meconopsis horridula* Hook. f. & Thomson | prickly blue poppy | Temperate Asian (*Fan et al., 2015*) | Sedative, analgesic (*Fan et al., 2015*) | JX087717 |
| Papaveraceae (Ranunculales) | *Papaver somniferum* L. | opium poppy | African and Middle Eastern (*Vetulani, 2001*) | Hallucinogen, sedative, analgesic, aphrodisiac (*Rätsch, 2005*) | KU204905 |
| Passifloraceae (Malpighiales) | *Passiflora* spp. | passion flower | Native American (*Rätsch, 2005*) | Antianxiety, sedative (*Heinrich et al., 2012*) | HQ900864 |
| Passifloraceae (Malpighiales) | *Turnera diffusa* Willd. ex Schult. | damiana | Native American (*Rätsch, 2005*) | Stimulant, antianxiety, aphrodisiac (*Rätsch, 2005*) | JQ593109 |
| Phytolaccaceae (Caryophyllales) | *Phytolacca acinosa* Roxb. | pokeweed | Temperate Asian (*Rätsch, 2005*) | Hallucinogen (*Rätsch, 2005*) | HM850257 |
| Piperaceae (Piperales) | *Arundo donax* L. | giant reed | African and Middle Eastern; Native American (*Rätsch, 2005*) | Hallucinogen (*Rätsch, 2005*) | U13226 |
| Piperaceae (Piperales) | *Piper* spp. | pepper, kava | Native American, Indomalayan, Australasian (*Rätsch, 2005*) | Stimulant, antianxiety, sedative, analgesic, aphrodisiac (*Rätsch, 2005*) | AY032642 |

Alrashedy and Molina (2016), *PeerJ*, DOI 10.7717/peerj.2546

| Family (Order) | Accepted binomial name | Common name | Indigenous psychoactive culture | Mechanism of action | Genbank numbers |
|---|---|---|---|---|---|
| Plantaginaceae (Lamiales) | *Bacopa monnieri* (L.) Wettst. | brahmi | Indomalayan (*Shinomol, Muralidhara & Bharath, 2011*) | Antianxiety, aphrodisiac (*Shinomol, Muralidhara & Bharath, 2011*) | KJ773301 |
| Poaceae (Poales) | *Lolium temulentum* L. | bearded darnel | African and Middle Eastern (*Rätsch, 2005*) | Hallucinogen (*Rätsch, 2005*) | KM538829 |
| Ranunculaceae (Ranunculales) | *Aconitum* spp. | monkshood | European, Indomalayan, Temperate Asian (*Rätsch, 2005*) | Hallucinogen, analgesic, aphrodisiac (*Rätsch, 2005*) | EU053898 |
| Ranunculaceae (Ranunculales) | *Hydrastis canadensis* L. | goldenseal | Native American (*Foster & Duke, 2000*) | Stimulant, sedative, analgesic (*O'Mahony, 2010*) | L75849 |
| Rubiaceae (Gentianales) | *Catunaregam nilotica* (Stapf) Tirveng. (=*Randia nilotica* Stapf) | chibra | Africa and Middle Eastern (*Danjuma et al., 2014*) | Antianxiety, antidepressant (*Danjuma et al., 2014*) | AJ286700 |
| Rubiaceae (Gentianales) | *Coffea arabica* L. | coffee | African and Middle Eastern (*Rätsch, 2005*) | Stimulant (*Rätsch, 2005*) | EF044213 |
| Rubiaceae (Gentianales) | *Corynanthe* spp. | pamprama | African and Middle Eastern (*Rätsch, 2005*) | Stimulant and aphrodisiac (*Rätsch, 2005*) | AJ346977 |
| Rubiaceae (Gentianales) | *Mitragyna speciosa* (Korth.) Havil | kratom | Indomalaya (*Idayu et al., 2011*; *Suhaimi et al., 2016*) | Stimulant, analgesic, sedative (*Rätsch, 2005*; *Suhaimi et al., 2016*) | AJ346988 |
| Rubiaceae (Gentianales) | *Pausinystalia johimbe* (K.Schum.) Pierre ex Beille | yohimbe | African and Middle Eastern (*Rätsch, 2005*) | Hallucinogen, stimulant, antidepressant, aphrodisiac (*Rätsch, 2005*) | AJ346998 |
| Rubiaceae (Gentianales) | *Psychotria* spp. | chacruna | Native American (*Rätsch, 2005*) | Hallucinogen, sedative, analgesic (*Rätsch, 2005*) | KJ805654 |
| Santalaceae (Santalales) | *Santalum murrayanum* C.A Gardner | sandalwood | Australasian (*Rätsch, 2005*) | Sedative (*Rätsch, 2005*) | L26077 |
| Sapindaceae (Sapindales) | *Paullinia* spp. | guarana | Native American (*McClatchey et al., 2009*) | Stimulant (*McClatchey et al., 2009*) | AY724365 |
| Solanaceae (Solanales) | *Atropa belladonna* L. | belladonna | European (*Schultes, 1976*) | Hallucinogen, stimulant, sedative, aphrodisiac (*Rätsch, 2005*) | AJ316582 |
| Solanaceae (Solanales) | *Brugmansia* spp. | angel's trumpet | Native American (*Rätsch, 2005*) | Hallucinogen, sedative, aphrodisiac (*Rätsch, 2005*) | HM849829 |

**Table 1** (*continued*)

| Family (Order) | Accepted binomial name | Common name | Indigenous psychoactive culture | Mechanism of action | Genbank numbers |
|---|---|---|---|---|---|
| Solanaceae (Solanales) | *Brunfelsia* spp. | raintree | Native American (*Rätsch, 2005*) | Hallucinogen, analgesic (*Rätsch, 2005*) | AY206720 |
| Solanaceae (Solanales) | *Cestrum* spp. | flowering jessamine | Native American (*Rätsch, 2005*) | Hallucinogen, sedative, analgesic (*Rätsch, 2005*) | JX572398 |
| Solanaceae (Solanales) | *Datura* spp. | toloache | Native American, Indomalayan, European (*Rätsch, 2005*) | Hallucinogen, sedative, analgesic, aphrodisiac (*Rätsch, 2005*) | JX996059 |
| Solanaceae (Solanales) | *Duboisia* spp. | pituri | Australasian (*Rätsch, 2005*) | Hallucinogen, stimulant, aphrodisiac (*Rätsch, 2005*) | KM895868 |
| Solanaceae (Solanales) | *Hyoscyamus* spp. | Henbane | European (*Rätsch, 2005*) | Hallucinogen. sedative (*Rätsch, 2005*) | KF248009 |
| Solanaceae (Solanales) | *Iochroma fuchsioides* (Bonpl.) Miers | yas | Native American (*Rätsch, 2005*) | Sedative (*Rätsch, 2005*) | KU310432 |
| Solanaceae (Solanales) | *Mandragora* spp. | mandrake | European, African and Middle Eastern (*Rätsch, 2005*; *Sayin, 2014*) | Hallucinogen, sedative, analgesic, aphrodisiac (*Rätsch, 2005*; *Sayin, 2014*) | U08614 |
| Solanaceae (Solanales) | *Nicotiana* spp. | tobacco | Native American, Australasian (*Vetulani, 2001*; *Rätsch, 2005*) | Stimulant, antianxiety (*Rätsch, 2005*) | KU199713 |
| Solanaceae (Solanales) | *Petunia violacea* Lindl. | shanin | Native American (*Schultes, 1976*) | Hallucinogen (*Schultes, 1976*) | HQ384915 |
| Solanaceae (Solanales) | *Physalis* spp. | groundcherry | Native American (*Rätsch, 2005*) | Sedative, analgesic (*Rätsch, 2005*) | KP295964 |
| Solanaceae (Solanales) | *Scopolia carniolica* Jacq. | scopolia | European (*Rätsch, 2005*) | Hallucinogen, sedative, aphrodisiac (*Rätsch, 2005*) | HQ216145 |
| Solanaceae (Solanales) | *Solandra* spp. | arbol del viento | Native American (*Knab, 1977*; *Rätsch, 2005*) | Hallucinogen, aphrodisiac (*Knab, 1977*; *Rätsch, 2005*) | U08620 |
| Solanaceae (Solanales) | *Solanum* spp. | nightshade | European, Native American (*Rätsch, 2005*) | Sedative, analgesic (*Rätsch, 2005*) | KC535803 |
| Solanaceae (Solanales) | *Withania somnifera* (L.) Dunal | ashwagandha | Indomalayan (*Rätsch, 2005*) | Sedative, aphrodisiac (*Rätsch, 2005*) | FJ914179 |
| Theaceae (Ericales) | *Camellia sinensis* (L.) Kuntze | tea | Temperate Asian (*Rätsch, 2005*) | Stimulant, aphrodisiac (*Rätsch, 2005*) | EU053898 |
| Urticaceae (Rosales) | *Urtica urens* L. | nettle | African and Middle Eastern (*Doukkali et al., 2015*) | Hallucinogen, antianxiety, sedative (*O'Mahony, 2010*; *Doukkali et al., 2015*) | KM361027 |

of all known plant species that is maintained by the Royal Botanic Gardens and the Missouri Botanical Garden. The psychoactive uses of each plant were categorized as follows: hallucinogen, sedative (=narcotic/hynotic), stimulant, anxiolytic (=relaxant), and antidepressant. As psychotropic plants may also exert analgesia and/or aphrodisiac effects, these effects were determined for each plant in addition to their original psychoactive use. Multiple effects based on literature were not uncommon. Thus, plants were assigned multiple psychoactive attributes, if applicable. For congeneric taxa, uses for each species were all noted.

The 126 psychoactive plant taxa were categorized according to the ethnic groups they were associated with: Native American (including North, Central and South America, 49 genera), European (15), Temperate Asian (including China, Russia, 10), Middle Eastern and African (19), Indomalayan (including India and Southeast Asia, 10), Australasia (including Australia, New Guinea, New Zealand, Pacific Islands, 4). Taxa with traditional psychoactive uses in at least two of these groups were designated multi-cultural (19). The uses of the plants were based on the originating indigenous cultures. For example, harmal, *Peganum harmala* (Nitrariaceae), is native in the Mediterranean (Europe), but it was used as a stimulant in the Middle East and in Africa, so harmal was assigned to the latter. Guava, *Psidium guajava* (Myrtaceae), is native to tropical America, but was only used as psychoactive in Africa (*Rätsch, 2005*). *Argyreia nervosa* (=*A. speciosa*), though of Indian origin, is considered multi-cultural here. It has been used in Ayurvedic medicine as an analgesic and aphrodisiac (*Galani, Patel & Patel, 2010*), but Hawaiians (Australasia) have been using it as alternative to marijuana (*Rätsch, 2005*). Cultural designations for each plant were all noted, with overlapping origins, if applicable, indicated.

To construct the phylogeny, the sequence of *rbcL* (the gene that codes for the photosynthetic enzyme rubisco; *Clegg, 1993*) for each psychoactive plant taxon was obtained from the GenBank database (http://www.ncbi.nlm.nih.gov/genbank) using BLASTN (*e*-value = 0, query coverage >50%; *Altschul et al., 1990*). If there are multiple species within the genus, only the genus name was indicated. The *rbcL* sequences were not available in GenBank for the following species: *Calea ternifolia, Calliandra anomala, Crocus sativus, Horsfieldia australiana, Iochroma fuchsioides, Juniperus recurva, Justicia pectoralis, Lactuca virosa, Ledum palustre, Lonchocarpus violaceus, Nymphaea ampla, Pachycerus pectenaboriginum, Psychotria viridis, Ptychopetalum olacoides, Psidium guajava, Rhynchosia pyramidalis, Sassafras albidum, Sceletium tortuosum, Tanaecium nocturnum, Tilia tomentosa, Urtica urens, Veratrum album,* and *Virola elongata*. In these cases, the *rbcL* sequence for any species within the corresponding genus was downloaded instead.

The *rbcL* sequences of the psychoactive plants were aligned using default parameters in MAFFT v.7 (*Katoh & Standley, 2013*). PhyML (*Guindon & Gascuel, 2003*) was utilized to reconstruct the phylogeny applying the general time reversible (GTR) DNA model (*Tavaré, 1986*) with aLRT (approximate likelihood ratio test) Shimodaira-Hasegawa-like (SH-like) branch support (*Simmons & Norton, 2014*) and 100 bootstrap replicates. ITOL (Interactive Tree of Life, http://itol.embl.de), a web-based tool used for the display and manipulation of phylogenetic trees (*Letunic & Bork, 2006*), was used to highlight and map the traits in Table 1 (indigenous culture, psychoactive uses). Affected neurotransmitter

(NT) systems (Table 2) for the main psychoactive families were also added to the phylogeny. Cosmetic editing of the ITOL results was completed in Adobe Illustrator CS4.

## RESULTS

The 126 psychoactive seed plant taxa belong to 56 families and 31 orders (Table 1) and together comprise 1.6% of the total generic diversity for these families. The phylogeny reflects expected relationships (*The Angiosperm Phylogeny Group, 2016*). Within eudicots there seems to be cultural bias of psychotropic use toward asterid members (61) vs. rosids (31). Nonetheless, the scattered distribution of psychoactive taxa throughout the angiosperm phylogeny suggests that psychoactive phytochemicals have evolved multiple times throughout angiosperm evolution. However, certain families are more diverse with at least 3 or more genera: Myristicaceae, Papaveraceae, Malvaceae, Fabaceae, Cactaceae, Asteraceae, Convolvulaceae, Solanaceae, Lamiaceae, Rubiaceae, Apocynaceae. However, psychoactive diversity within these families may be positively correlated with the family's generic diversity. To test this, a Pearson's product moment correlation coefficient was calculated to test the relationship between the number of psychoactive genera in our study versus the generic diversity of each family (from *Christenhusz & Byng, 2016*). Taxonomically diverse families like Asteraceae and Rubiaceae (>500 genera each) did not always have proportionally higher number of psychoactive genera with the correlation coefficient very weakly positive ($r = 0.004$). However, Myristicaceae (4 psychoactive genera out of 21 total), Papaveraceae (4/42), Cactaceae (5/127), Convolvulaceae (4/53), Solanaceae (16/100), Lamiaceae (8/241), Apocynaceae (7/366) have a disproportionate number (>1.6%) of their family's generic diversity psychoactive. We focused on the neurotransmitter systems affected by psychotropic members of these families as well as psychoactive members in the inherently diverse families of Fabaceae, Malvaceae, Rubiaceae, and Asteraceae (Fig. 1).

Unrelated families may exert similar psychoactive effects (Fig. 1). Cactaceae, Fabaceae, Myristicaceae, Convolvulaceae, and Solanaceae are mainly hallucinogens, though they are unrelated. Of the five cultural groups, Native Americans have traditionally used the most psychoactives (49/126) with predilection for hallucinogens (Fig. 2) in Cactaceae, Fabaceae, Convolvulaceae. These families mainly work as serotonin receptor agonists (Fig. 1; Table 2), the same mechanism as hallucinogenic Myristicaceae that has been used in Australasia and Indomalaya. Members of Solanaceae have also been used as hallucinogens, predominantly by Native Americans and Europeans, but act via a different mechanism—as acetylcholine antagonists. Hallucinogenic asterids are also often used as aphrodisiacs (16/30 = 53% vs. 4/18 = 22% hallucinogenic rosids).

The unrelated Papaveraceae and Lamiaceae similarly show sedative/narcotic qualities, another popular psychoactive effect among different cultural groups (Fig. 2). However, they affect different neurotransmitter systems with Papaveraceae working mainly as opioid receptor agonists. Lamiaceae work as receptor agonists of gamma-amino butyric acid (GABA), which also mediates the family's anxiolytic effects. Psychoactive members of these families also tend to exhibit analgesic effects.

Plants with anxiolytic and antidepressant properties are relatively sparse (Figs. 1 and 2), with Europeans showing slightly increased use of these plants. Members of Apocynaceae

**Table 2 Main psychoactive families (cf. Fig. 1), their primary psychoactive effect, suspected phytochemical constituents producing the effect, and the primary neurotransmitter (NT) systems potentially affected.** "±" refers to the activation (receptor agonist) and inhibition (receptor antagonist), respectively, of certain NT receptors by the psychoactive substance.

| Family | Main psychoactive effect | Active phytochemicals | Neurotransmitter systems affected |
| --- | --- | --- | --- |
| Apocynaceae | Antidepressant | Indole alkaloids, e.g., ibogaine, rauwolscine, reserpine, yohimbine (*Spinella, 2001*; *Polya, 2003*; *Rätsch, 2005*; *Pratchayasakul et al., 2008*; *Sayin, 2014*; *Cardoso et al., 2015*) | Serotonin (+), dopamine (+), noradrenaline (+) (*Wells, Lopez & Tanaka, 1999*; *Spinella, 2001*; *Polya, 2003*; *Grundmann et al., 2007*; *Arulmozhi et al., 2012*; *Zheng, Fan & Liu, 2013*; *Sayin, 2014*; *Cardoso et al., 2015*) (except reserpine but other indole alkaloids may counteract its effects (*Polya, 2003*) |
| Asteraceae | Hallucinogen, aphrodisiac | Sesquiterpene lactones (*Rätsch, 2005*; *Sayin, 2014*) | Unknown mechanisms for various sesquiterpene lactones (*Chadwick et al., 2013*) |
| Cactaceae | hallucinogen | Phenethylamine alkaloids, e.g., hordenine, mescaline, pectenine (*Rätsch, 2005*; *Sayin, 2014*) | Serotonin (+) (*Polya, 2003*) |
| Convolvulaceae | hallucinogen | Ergot indole alkaloids (*Rätsch, 2005*; *McClatchey et al., 2009*) | Serotonin (+) (*Polya, 2003*; *Kennedy, 2014*) |
| Fabaceae | Hallucinogen | Indole alkaloids, e.g., bufotenin, DMT; tryptamines (*Polya, 2003*; *Wink, 2003*; *Halpern, 2004*; *Rätsch, 2005*) | Serotonin (+) |
| Lamiaceae | Anxiolytic, sedative, analgesic | Terpenoids e.g., baicalin, linalool, labdane, rosmarinic acid, salvinorin A, wogonin, etc. (*Lis-Balchin & Hart, 1999*; *Awad et al., 2003*; *Awad et al., 2009*; *Polya, 2003*; *Wink, 2003*; *Heinrich et al., 2012*); leonurine alkaloid (*Rauwald et al., 2015*) | GABA (+) (*Awad et al., 2003*; *Awad et al., 2009*; *Hajhashemi, Ghannadi & Sharif, 2003*; *Shi et al., 2014*; *Rauwald et al., 2015*) |
| Malvaceae | Stimulant | Xanthine alkaloids, e.g., caffeine, theobromine (in *Cola*, *Theobroma*; *Rätsch, 2005*; *McClatchey et al., 2009*); phenethylamine ephedrine (in *Sida*; *Prakash, Varma & Ghosal, 1981*) | Adenosine (−) by xanthine alkaloids (*Polya, 2003*; *McClatchey et al., 2009*); adrenaline (+) by ephedrine (*Polya, 2003*) |
| Myristicaceae | Hallucinogen | DMT (indole alkaloid in *Virola*); phenylpropene e.g., myristicin, elemicine, safrole (*Polya, 2003*; *Rätsch, 2005*) | Serotonin (+) (*Spinella, 2001*; *Polya, 2003*) |
| Papaveraceae | Hallucinogen | Isoquinoline alkaloids, e.g., codeine; morphine; reticuline; thebaine (*Polya, 2003*; *Heinrich et al., 2012*; *Fedurco et al., 2015*; *Shang et al., 2015*) | Opioid (+) (*Rolland et al., 1991*; *Polya, 2003*; *Shang et al., 2015*) |
| Rubiaceae | Stimulant | caffeine (xanthine alkaloid in *Coffea*; *Polya, 2003*); indole alkaloids in others, e.g., corynanthine, mitragynine, yohimbine (indole alkaloid; *Polya, 2003*; *Rätsch, 2005*; *Suhaimi et al., 2016*) | Adenosine (−) by xanthine alkaloids (*Polya, 2003*; *McClatchey et al., 2009*); adrenaline (+) and serotonin (+) by indole alkaloids (*Polya, 2003*) |
| Solanaceae | Hallucinogen, sedative, | Tropane alkaloids, e.g., atropine, hyoscyamine, scopolamine (*Polya, 2003*; *Wink, 2003*; *Rätsch, 2005*) | Acetylcholine (−) (*Polya, 2003*) |

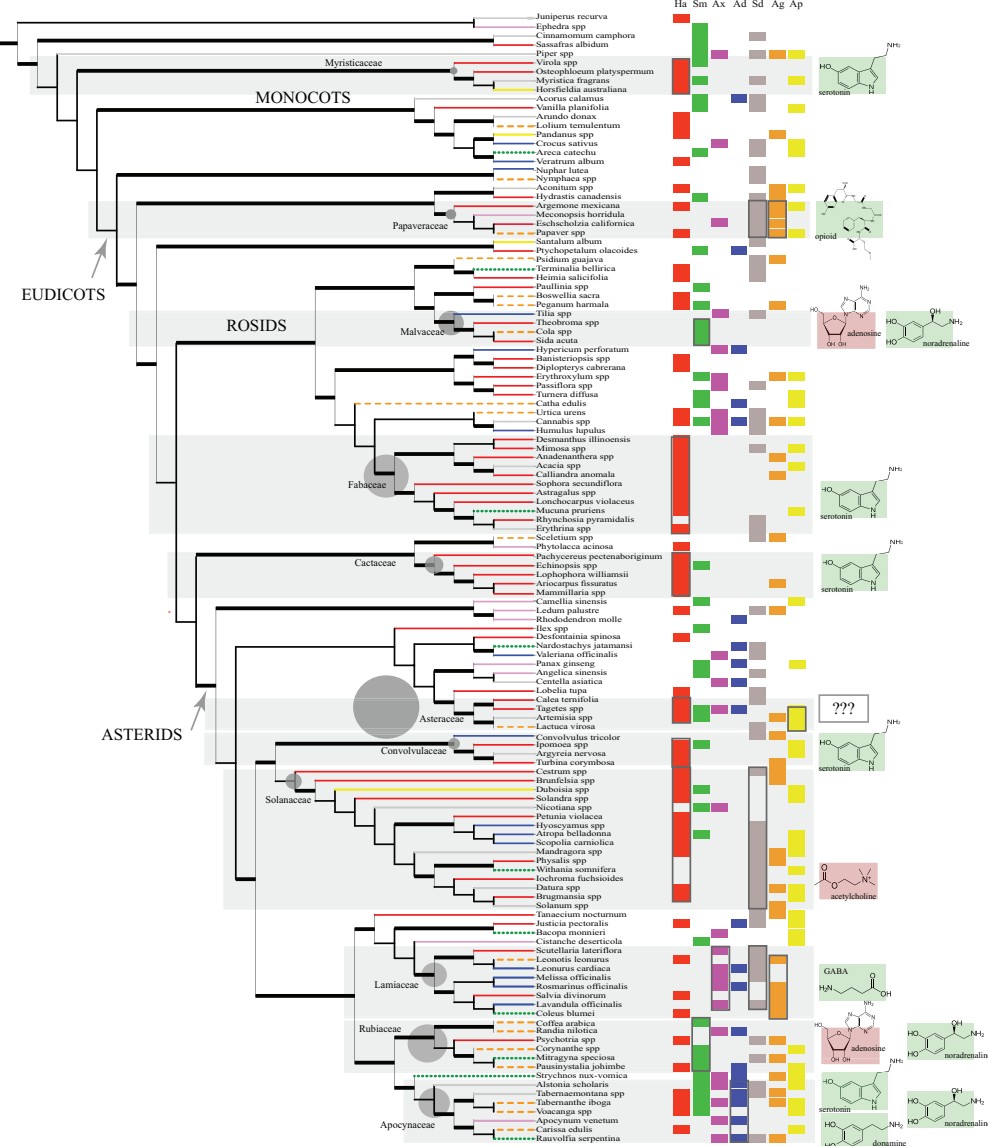

**Figure 1** **The phylogeny (cladogram) of traditionally used psychoactive plant taxa.** The phylogeny conforms to expected groupings (APG IV, 2016). The 11 main plant families are highlighted (top to bottom): Myristicaceae, Papaveraceae, Malvaceae, Fabaceae, Cactaceae, Asteraceae, Convolvulaceae, Solanaceae, Lamiaceae, Rubiaceae, Apocynaceae. Grey circles next to their family names are proportional to total generic diversity within the family with lowest count for Myristicaceae (21 genera), and highest with 1623 genera for Asteraceae (*Christenhusz & Byng, 2016*). Branches are coded according to the different cultures (Native American: red solid line; Middle Eastern and African: orange dashed line; European: blue solid line; Indomalayan: green dotted line; Temperate Asia: pink solid line, Australasia: yellow solid line; Multicultural: grey solid line). Branches in bold represent bootstrap node support >50% and SH-like branch support >0.9. Psychoactive uses were overlain next to taxon names in columns (Ha, hallucinogen; Sm, stimulant; Ax, anxiolytic; Ad, antidepressant; Sd, sedative; Ag, analgesic; Ap, aphrodisiac; along with the primary neurotransmitters affected by the phytochemical/s exerting the dominant psychoactive effect (delineated with boxes; cf. Table 2). Shaded plant families with phytochemicals that activate certain neurotransmitter systems (e.g., receptor agonists) show the neurotransmitter/s involved with green (bright) background; phytochemicals with inhibitory effects to the NT have red (dark) background. In Asteraceae, neuropharmacology is unclear (???).

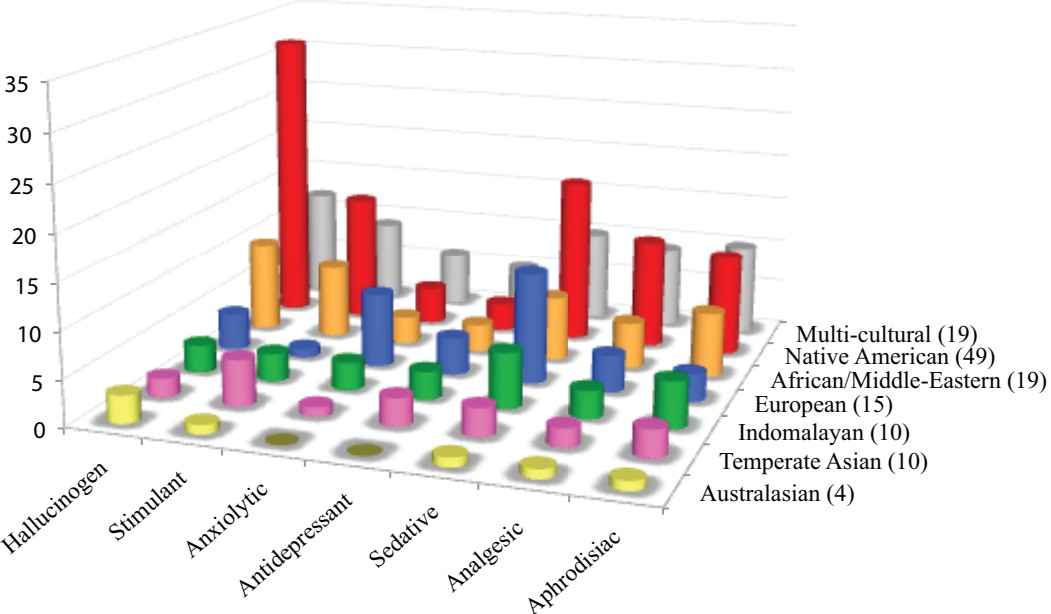

**Figure 2 Cultural distribution of psychoactive applications.** Psychoactive plants were categorized according to cultural affiliation and psychoactive uses. Each row shows the distribution of psychoactive uses for plants within a cultural group. Of the 126 psychoactive plant genera, more than half of the plants are used as hallucinogens mostly by Native Americans. Plants with sedative/narcotic qualities are also commonly sought after. Plants with anxiolytic and antidepressant effects are the least popular among different cultures.

and Rubiaceae that show an antidepressant effect facilitate this effect by increasing synaptic levels of monoamine neurotransmitters (serotonin, dopamine, noradrenaline; Fig. 1 and Table 2). In contrast, plants with stimulating effects are numerous and randomly distributed throughout the phylogeny, exhibiting varying mechanisms of action (see Malvaceae and Rubiaceae, Fig. 1 and Table 2).

## DISCUSSION

The phylogenetic distribution of psychoactive plants shows multiple evolutionary origins and provides evidence for the adaptive benefit of phytochemicals that are psychoactive in animals. It has been hypothesized that mammals may have sought plants with these phytochemicals that were chemically similar to endogenous neurotransmitters to augment their nutrition, as well as to facilitate survival, alleviating pain and hunger (*Sullivan & Hagen, 2002*). Whether this phylogenetic distribution, showing multiple independent origins of psychoactive plants, is due to co-evolutionary mutualism with animals remains to be tested. However, it is clear that certain psychoactive effects are concentrated in certain groups, which demonstrates that psychoactive phytochemicals are phylogenetically clustered. Phylogenetic clustering of certain secondary metabolites (*Wink, 2003*; *Wink et al., 2010*; *Wink, 2013*) and of medicinal traits (*Saslis-Lagoudakis et al., 2012*; *Saslis-Lagoudakis et al., 2015*; *Xavier & Molina, 2016*) have also been revealed in other studies.

In the phylogeny, 11 of 56 plant families have more psychoactive genera (three or more) compared to others. Accounting for these families' total generic diversity shows that Myristicaceae, Papaveraceae, Cactaceae, Convolvulaceae, Solanaceae, Lamiaceae, and Apocynaceae have a disproportionate number of psychoactive genera. The psychoactive diversity of the other families, Fabaceae, Malvaceae, Asteraceae, and Rubiaceae, may be an artifact of their overall higher generic diversity. Nonetheless, we see a pattern where these plant families are being used for similar psychoactive applications by different cultures, a pattern of cultural convergence (*Xavier & Molina, 2016*) with bias, interestingly, for plants with hallucinogenic and sedative/narcotic potential.

## Pharmacology of hallucinogenic plants

The use of hallucinogens is widespread in cultures which assigned positive meaning to the experienced altered state of consciousness, such as allowing the user access to the spiritual world (*Júnior et al., 2015*). Hallucinogens used in divination and religious healing (i.e., entheogens) may have played a significant role in human evolution (*Schultes, Hofmann & Rätsch, 2001*). Native Americans prolifically used hallucinogens, but hallucinogenic use seems to be lower in temperate Asia. Increased hallucinogenic use among indigenous peoples of Brazil (South America) was also reported by *Rodrigues & Carlini (2006)*.

In our study we find hallucinogenic plants in Myristicaceae, Fabaceae, Cactaceae, and Convolvulaceae mainly acting as serotonin receptor agonists, a case of mechanistic convergence where unrelated families exert the same psychoactive effect by affecting identical neurotransmitter systems. Mescaline is the serotonergic chemical in Cactaceae, while DMT (N,N-dimethyltryptamine) and bufotenin (*Polya, 2003*) have the same effect and evolved independently in hallucinogenic taxa in Fabaceae (*Wink, 2013*). Serotonin itself occurs in fabaceous *Mucuna pruriens* (*Polya, 2003*), a hallucinogen and aphrodisiac in Ayurvedic medicine (*Lampariello et al., 2012*). DMT also exists in *Virola* of the unrelated Myristicaceae (*Polya, 2003*), and the alkaloid, elemicine, in confamilial *Myristica fragrans* transforms into a mescaline-like molecule (*Rätsch, 2005*).

The unrelated Convolvulaceae exerts hallucinogenic effects possibly through its ergot alkaloids that work also as serotonin receptor agonists (*Polya, 2003*; *Kennedy, 2014*). Yet interestingly, these ergot alkaloids originate from ascomycetous symbiotic fungi (*Beaulieu et al., 2013*). Though endophytic fungi can produce some active metabolites originally attributed to plants (*Wink, 2008*; *Wink et al., 2010*; *Nicoletti & Fiorentino, 2015*), which may confound interpretation of the phylogeny, this was not the case, so far, for the other main psychoactive families in our study. On the other hand, hallucinogenic taxa in the closely related Solanaceae work on a different mechanism. Its tropane alkaloids such as scopolamine and atropine act as muscarinic receptor antagonists, inhibiting acetylcholine transmission (*Spinella, 2001*). Interestingly, in another asterid member, *Salvia divinorum* (Lamiaceae), the diterpene, salvinorin A, possibly works as a hallucinogen through its action on specific opioid receptors (kappa) (*Willmore-Fordham et al., 2007*), the same receptor modulated by the alkaloid ibogaine in hallucinogenic *Tabernanthe iboga* (Apocynaceae;

*Spinella, 2001*). Various unrelated taxa seemingly achieve their hallucinogenic effects by modulating serotonin, acetylcholine, and/or endogenous opioids.

It is interesting that in many hallucinogenic asterids, aphrodisiac effects are quite common (see Asteraceae, Solanaceae, Apocynaceae). In members of Solanaceae this effect may be due to dopamine increase from cholinergic antagonism (*Spinella, 2001*). Dopamine is important in sexual arousal and orgasm (*Krüger, Hartmann & Schedlowski, 2005*). This neurotransmitter is also modulated by ibogaine in *T. iboga* (*Wells, Lopez & Tanaka, 1999*), which is also traditionally used as an aphrodisiac along with other Apocynaceae members. In another asterid family, Asteraceae, it is not clear which of its phytochemical constituents produce psychoactive effects, except perhaps for wormwood (*Artemisia* spp.) wherein the monoterpenoid, thujone, antagonizes the main inhibitory neurotransmitter, gamma-aminobutyric acid (GABA), resulting in its stimulant, almost convulsant, effects (*Höld et al., 2000*). However, the great diversity of sesquiterpene lactones prevalent in the family (*Chadwick et al., 2013*) are likely implicated in its hallucinogenic and aphrodisiac potential (Fig. 1 and Table 2). These findings motivate further research into these asterid families as new therapeutics for sexual dysfunction.

## Pharmacology of plants with sedative and analgesic effects

Dr. WE Dixon, well-known British pharmacologist of his time, once commented that narcotic indulgences reflect the sad paradox that humans seemed to get their "chief pleasures in life by escaping out of life" (*Narcotic Plants, 1928*: 252). There may be truth to this as narcotic/sedative effects were commonly sought for by various cultures, second to hallucinogens, with members of Papaveraceae and Lamiaceae traditionally used for this purpose. Opium poppy of Papaveraceae has long been known to ancient Greeks and Sumerians and is considered one of the most important medicinal plants in history. Its opium latex is the source of >30 alkaloids including morphine and codeine, which bind to opioid receptors, promoting sedation and analgesia (*Heinrich et al., 2012*). Though there are other members of Papaveraceae that have been used by Asians and Native Americans for sedation and pain relief (*Rolland et al., 1991*; *Brahmachari, Gorai & Roy, 2013*; *Shang et al., 2015*), the substances responsible for their effects are not well characterized as in opium poppy, but it is possible that their effects are also mediated via opioid receptors (*Shang et al., 2015*) and at least in *Eschscholzia californica* (California poppy) via the GABAergic system (*Fedurco et al., 2015*).

In asterids, sedation is produced by members of Solanaceae and Lamiaceae possibly via different pathways. Tropane alkaloids in Solanaceae (*Wink, 2003*), particularly scopolamine, promote sedation through depression of the central nervous system resulting from anticholinergic activity (*Renner, Oertel & Kirch, 2005*). In Lamiaceae, this effect is mainly facilitated via the GABAergic pathway (*Shi et al., 2014*), with leonurine (*Rauwald et al., 2015*) and essential oil components (*Lisbalchin & Hart, 1999*; *Wink, 2003*; *Awad et al., 2009*; *Shi et al., 2014*; *Ferlemi et al., 2015*) as the primary chemicals that increase GABA. Coincidentally, Lamiaceae members also possess analgesic effects, but the pharmacology is unclear (*Hajhashemi, Ghannadi & Sharif, 2003*; *Dobetsberger & Buchbauer, 2011*) and may reflect the antinociceptive properties of activation of GABA

receptors (*Enna & McCarson, 2006*). *Salvia divinorum* however, does not contain essential oils (*Rätsch, 2005*), but has been pharmacologically shown to exert analgesic quality through activation of the same opioid receptors (kappa) implicated in its hallucinogenic effect (*Willmore-Fordham et al., 2007*), a mechanism different from the other Lamiaceae species here. Some members of the distantly related Rubiaceae, including *Psychotria colorata* (*Elisabetsky et al., 1995*) and *Mitragyna speciosa* (*Suhaimi et al., 2016*), have also shown similar opiate-like antinociceptive properties, confirming their traditional uses. Repeated evolution of phytochemicals with affinity for animal opioid receptors may imply some adaptive benefit to plants.

## Pharmacology of plants with anxiolytic and antidepressant effects

The relatively sparse distribution of anxiolytic and antidepressant plants in the phylogeny compared to hallucinogens and sedatives, suggests that there is less cultural utility for plants with these psychoactive properties. In the US there is a cultural aspect to the pathogenesis of anxiety and depression with minority groups reporting lower incidence compared to whites (*Hofmann, Asnaani & Hinton, 2010*). The definition itself of depression is wrought with Western assumptions of individual happiness, which is in contrast to other cultures' view of happiness arising from social interdependence (*Chentsova-Dutton, Ryder & Tsai, 2014*). This may explain why these psychoactive uses were less prevalent compared to hallucinogenic, stimulant and sedative applications.

Sedative members of Lamiaceae often possess anxiolytic qualities (Fig. 1), and this is probably due to overlapping effects on GABA (*Tallman et al., 2002*). Phytol, an alcohol in essential oils (*Costa et al., 2014*) has been shown to increase GABA. Rosmarinic acid in rosemary *(R. officinalis)* and lemon balm *(M. officinalis)*, both Lamiaceae, also works as GABA transaminase inhibitor preventing GABA catabolism (*Awad et al., 2009*).

In members of Apocynaceae and Rubiaceae (Gentianales) that show anxiolytic and antidepressant effects, another mechanism may be involved. *Rauvolfia serpentina* (Apocynaceae) is used in Ayurvedic medicine to treat depression (*Mamedov, 2005*). In Africa, the confamilial *T. iboga* is used as a stimulant to combat fatigue and hunger, but may have potential in easing depressive symptoms (*Nigg & Seigler , 1992*). *Pausinystalia yohimbe* (Rubiaceae) has stimulating effects on the nervous system and has been used to increase libido by men in central Africa (*Rätsch, 2005*). The confamilial *M. speciosa* has also been used as stimulant to counteract fatigue and increase endurance for work in Southeast Asia (*Idayu et al., 2011*). The main chemical constituents of these closely related families are indole alkaloids that generally increase synaptic levels of the monoamine neurotransmitters, serotonin, dopamine and noradrenaline by various mechanisms including inhibition of transport and reuptake (*Wells, Lopez & Tanaka, 1999*; *Zheng, Fan & Liu, 2013*; *Kennedy, 2014*). The unrelated but popular herbal antidepressant, St. John's wort (*Hypericum perforatum*, Hypericaceae; *Spinella, 2001*), as well as pharmaceutical antidepressants, produces its effects (*Feighner, 1999*) via the same mechanism of reuptake inhibition.

Monoamine transport inhibitors may be rife in Apocynaceae (or Gentianales). In her ethnopharmacological studies of plants from South Africa, *Jäger (2015)* also discovered two other Apocynaceae species that exhibited high affinity to the serotonin transporter.

Interestingly, these plants were also being used by traditional healers to treat those who were "being put down by the spirits." A primary side effect of many conventional antidepressants is sexual dysfunction (*Higgins, Nash & Lynch, 2010*), which seems to contradict the aphrodisiac effect exhibited by *T. iboga* and *P. yohimbe*, in addition to their antidepressant effects. This suggests that members of Gentianales may be exploited as novel pharmaceuticals for depression without the known side effects of sexual dysfunction.

## Pharmacology of plants with stimulating effects

Plants traditionally used as stimulants are numerous and scattered throughout the phylogeny, indicating that stimulant phytochemicals have evolved multiple times independently in different lineages and may confer some evolutionary benefit. A few display paradoxical effects as both stimulating and sedating, such as marijuana (*Block et al., 1998*) and *M. speciosa* (*Rätsch, 2005*), which may be attributed to dosage, idiosyncrasies, or antagonistic phytochemicals.

Albeit belonging to diverse families, coffee (*Coffea arabica*, Rubiaceae), yerba mate (*Ilex paraguariensis*, Aquifoliaceae), kola (*Cola* spp., Malvaceae), tea (*Camellia sinensis*, Theaceae), and guarana (*Paullinia cupana*, Sapindaceae), all contain caffeine, a xanthine alkaloid, which acts as a stimulant through antagonism of adenosine receptors, interfering with the binding of the inhibitory endogenous adenosine (*Rätsch, 2005*). Yohimbe (*P. yohimbe*), though confamilial with coffee, contains the indole alkaloid, yohimbine, which binds to adrenergic and serotonin receptors (*Polya, 2003*), and is structurally and mechanistically similar to other stimulant alkaloids found in diverse plant groups such as ergot alkaloids in Convolvulaceae, ibogaine in *T. iboga* and *Voacanga* sp. (Apocynaceae), and harmaline in *Peganum harmala* (Nitrariaceae) (*Polya, 2003*).

Within the same family, particularly Solanaceae, contrasting effects and mechanisms may also be observed. Though many solanaceous members contain tropane alkaloids that work as anticholinergic hallucinogens with incapacitating effects, tobacco exerts stimulant activity through an opposite mechanism, with nicotine, a pyrrolidine alkaloid, promoting acetylcholine transmission. However, tropane alkaloids are not unique to Solanaceae. Cocaine, found in the unrelated *E. coca* (Erythroxylaceae), suggests that chemically similar alkaloids may evolve in divergent lineages (i.e., convergent evolution) or alternatively, certain metabolic pathways have been evolutionarily conserved throughout plant evolution and differential gene regulation is responsible for the expression of this pathway (*Wink, 2003*; *Wink, 2008*; *Wink et al., 2010*; *Weng, 2014*). These may account for the presence of ephedrine in the gymnosperm *Ephedra* spp. (Ephedraceae; *Polya, 2003*) and the unrelated angiosperms *Sida acuta* (Malvaceae; *Prakash, Varma & Ghosal, 1981*) and *Catha edulis* (Celastraceae; *Polya, 2003*). Ephedrine, a phenethylamine that mimics noradrenaline, stimulates the adrenergic receptor system, and thus the sympathetic nervous system responsible for the "fight-and-flight" response (*Polya, 2003*; *Rätsch, 2005*).

It is notable that, even within the same family, the stimulant phytochemicals are chemically diverse. This phylogenetic pattern may indicate that stimulant chemicals may be more evolutionarily labile than hallucinogenic and sedative phytochemicals that seem to be more phylogenetically conserved within the family. As to why this is begs further

inquiry, but hints at the evolutionary benefits of these chemically diverse plant psychoactive compounds that have evolved multiple times among seed plants, possibly with multifarious roles other than to function solely as allelochemicals.

## CONCLUSION

Phylogenetic analysis has demonstrated multiple evolutionary origins of traditionally used psychoactive plant groups. Whether this pattern is due to repeated co-evolutionary mutualism with animals remains to be tested. Psychoactive diversity of some highlighted families is probably due to the inherent elevated diversity in these families. However, other plant families have a disproportionate number of psychoactive genera, and their phytochemical and psychoactive traits show phylogenetic clustering, with different cultures converging on geographically-disparate members of these families for similar uses: Myristicaceae, Cactaceae, Convolvulaceae, and Solanaceae as hallucinogens; Papaveraceae, Lamiaceae for analgesia and sedation; Apocynaceae for antidepressant effects. In certain unrelated families with the same psychoactive effect, the same neurotransmitter systems were also affected, i.e., mechanistic convergence. However, this was not the case for plants with stimulant effects, where confamilial taxa possess chemically diverse stimulant alkaloids, and chemically similar stimulant alkaloids exist in diverse lineages. Endophytic fungi can also produce some active metabolites originally attributed to plants (*Wink, 2008*; *Wink et al., 2010*; *Nicoletti & Fiorentino, 2015*), and this should be considered when interpreting the phylogeny.

Though we may have missed other psychotropic taxa, our study still provides insight into the ethnobotanical origins of psychoactive plant use. The addition of these missing taxa may only serve to corroborate our conclusion of widespread human dependence on psychoactive plants and highlight other important psychoactive families and their pharmacology. The brain is perhaps the most complex domain of the human body (*Singer, 2007*), and therefore brain disorders are complex pathologies themselves (*Margineanu, 2016*). Ethnobotanical research on how various human cultures have exploited herbal therapy through time to treat neurological afflictions will continue to provide insight into the etiology of these diseases and the success of folkloric treatments. Yet, the astounding diversity of plant-based medicines may be better appreciated within an evolutionary context that can reveal phylogenetic patterns that may guide future drug discovery (*Saslis-Lagoudakis et al., 2012*; *Xavier & Molina, 2016*). Though chemically similar psychoactive chemicals may exist in phylogenetically unrelated lineages, suggesting convergent evolution or differential gene regulation of common metabolic pathways (*Wink, 2003*; *Wink, 2008*; *Wink et al., 2010*), the majority of traditionally used psychoactive plants generally display phylogenetic conservatism in phytochemistry and pharmacology, and may be explored as novel therapeutics for neurological disorders such as depression, anxiety, pain, insomnia and sexual dysfunction, reinforcing the potential of plant psychoactives as "springboards for psychotherapeutic drug discovery" (*McKenna, 1996*).

## ACKNOWLEDGEMENTS

This research was conceived as part of NA's MSc thesis, and we are grateful to the King Abdullah scholarship program (of Saudi Arabia) for sponsoring NA. We also thank NA's family and Michael Purugganan for various forms of support. We also thank Joseph Morin and Timothy Leslie for reviewing earlier drafts of this manuscript. We are equally grateful to the reviewers for their constructive comments.

### Funding

The authors received no funding for this work.

### Competing Interests

The authors declare there are no competing interests.

### Author Contributions

- Nashmiah Aid Alrashedy performed the experiments, analyzed the data, contributed reagents/materials/analysis tools, wrote the paper, prepared figures and/or tables, reviewed drafts of the paper.
- Jeanmaire Molina conceived and designed the experiments, performed the experiments, analyzed the data, contributed reagents/materials/analysis tools, wrote the paper, prepared figures and/or tables, reviewed drafts of the paper.

### Data Availability

The research in this article did not generate, collect or analyse any raw data or code.

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
