# Peer review of "The ethnobotany of psychoactive plant use: a phylogenetic perspective"

_PeerJ, doi:10.7717/peerj.2546_

## Round 0.1 · original submission · Minor Revisions

Dear authors

As you see, our reviewers made many comments. Please try to take care of them in the revision.

Regards

Michael Wink
Academic editor

Reviewer 1 ·

Basic reporting

No comments

Experimental design

No comments

Validity of the findings

No comments

Additional comments

This review paper gives a useful bird’s eye view of the global diversity of mind-altering plants and raises many interesting questions about these plants and their modes of action.
It is certainly an elegant contribution. The paper is well written (in very good English) and has almost no typing or spelling errors (I noted only a few, see below).
In my opinion the phylogenetic approach did not really yield any new insights. Hegnauer’s classical review of chemosystematics (and the work of many other authors since) has already highlighted the lack of congruence between chemical characters and taxonomic affinity and the fact that the same compounds have evolved independently in several different lineages of higher plants [e.g. macrocyclic pyrrolizidine alkaloids in Crotalaria (Fabaceae, i.e. rosales) and in Senecio (Asteraceae, asterales)]. I remain unconvinced that phylogenetic analysis is an appropriate tool at global level, especially when the sampling is somewhat incomplete and uneven for the southern Hemisphere. Why not simply use existing phylogenies and plot the chemical pattern onto the topology? The beautiful colour cladogram nevertheless is very impressive and provides a nice pictorial overview of the main patterns.
The absence of some critical taxa is noteworthy – perhaps because of an inherent Northern Hemisphere and American bias? There are very many unique mind-altering plants that may put a different perspective on some of the patterns and trends reported in this paper. The human mind remains one of the last frontiers of science and it is likely that it will be a strong research focus in the next millennium, just as relativity (thanks to Albert Einstein) and holism (thanks to Jan Smuts) have shaped scientific thinking in the current millennium (and continuing into the future). I am not sure if a phylogenetic approach will help much – basic ethnobotany seems to be a much more direct source of new research directions and new discoveries (see later on). In view of the general loss of traditional knowledge in rural parts of the word, basic ethnobotanical surveys seem to be a more productive approach for new discoveries (and to gain a more globally representative picture). Some mention of the general literature on the human mind and the future of neuropharmacology, and the fact that we have much to learn, may perhaps strengthen the paper.
I was disappointed not to see Boophone disticha (Amaryllidaceae), the main hallucinogen of the earliest of human lineages (the San people of southern Africa) in the species table. This is the chemical basis for the famous kia or trance dance, which shows (if nothing else) that shamanism and visits to “the other world” is not only prevalent in North America but also in Africa, at the very cultural roots of modern humans. Similar gaps exist for Dioscoreaceae (e.g. Dioscorea dreageana), Menispermaceae (Cissampelos species) and other African sedatives. The Mesembryanthemum or Sceletium species that are now revealing new avenues of research and possible new mechanisms of action are not called ice plants but rather canna or kougoed. Ice plants (i.e. the broad-leaved species of Mesembryanthemum) have no known neuro-activity. The traditional hypnotic masticatories are called canna, kanna or kougoed and refer to the group of short-lived perennial species with leaves that become sceletonised in summer (ice plants are annuals).
Another example of under-sampling in a global context is likely to be Madagascar, where more than 80% of the 13 000 odd species are endemic. Philippe Rasoanaivo and co-workers have, for example, showed that the neuroactivity of Neobeguea mahafalensis (Meliaceae) is due to three different phragmalin limonoids, each with a completely different (and probably as yet undescribed) neuropharmacology. The main compound (libiguin A) stimulates unprecedented, powerful and prolonged sexual activity in rats (with an initial lag phase of two to three days). Surprisingly, two structurally related limonoids have very different activities: dodoguin induces a moderately lasting sleep without affecting the GABA(A) –receptor, thus indicating a novel mechanism of action; gidraguin induces extreme non-provoked male-to-female aggression in rats. The important point here is that it is difficult to see how a phylogenetic approach could have predicted this interesting discovery (in Meliaceae of all places!).
Some minor points:
Line 111. Full stop is missing
Line 149. Typing error: “asutralian”
Line 189: five or 5? (potential inconsistency, depending on the journal style (also in 222)
Line 309: minority groups vs whites. I think this statement needs some thought. In many parts of the world, whites represent the minorities. Perhaps rather European or Western cultures vs other cultures?
Line 330: Full stop is missing.
Line 338: Replace “in South Africa” with “of South African plants”. This is a factual error – the researchers are not working in South Africa.

I hope these comments and perspectives are of some value to the authors and editors.

·

Basic reporting

All my comments in the last block

Experimental design

All my comments in the last block

Validity of the findings

All my comments in the last block

Additional comments

This work is mostly a review of most of our general knowledge of psychoactive plant use including an evolutionary perspective.
The text is very clear and easy to read and bibliography is up to date and most of time well chosen (even if surely alternative papers exist in some cases).
The organization of the text is may be a point to be improved. This work is mainly an exceptional review but the part result is finally rather small, mainly based on a phylogeny on a single gene where were indicated geographical origin and, in part, compounds involved in psychoactive uses.
I don’t know if a strategy should be to present this work as a review removing the part “results” or to keep this section with the originality of figure 1 and its information and to increase the part results by including figure 2. If so, I think that you may include in results an analysis taking in account not only cultural uses in relation with geography but also adding a point which may be is missing in your work: approximation of the first mention/dating of the various types of uses among the different cultures. Surely then you may solve one of the important point what you discussed only in part in your discussion: That Anxiolytic and Antidepressant compounds are extremely recent and are not corresponding to traditional medicine (I think that they appeared in Europe in the end of the 19th century? But maybe you’ve got some contradictory data?).

Sentence beginning line 42. Please include references.
Line 48, speaking of co-evolution is not correct in my mind, even if you modulate your text using “some kind of”. Co-evolution is in my mind a reciprocal selection force acting on genomes. Considering age of use by human being of this compounds, I doubt that you may speak of co-evolution sensu stricto.
Line 224: miss spelling of Papaveraceae

Line 307 see my comment in text which correspond to my suggestion above.

This work is in my mind of a very high level and I enjoyed to read it. I think that after few cosmetic changes/corrections, it will be worth to be published in PeerJ and will be extensively cited.

Reviewer 3 ·

Basic reporting

See below

Experimental design

See below

Validity of the findings

See below

Additional comments

The paper by Alrashedy and Molina focuses on a phylogenetic analysis of psychoactive plants. The phylogeny of the plants containing these metabolites based on the sequence of the rbcL gene is compared with the mostly erratic distribution of psychoactive metabolites and multiple evolutionary origins of psychoactive plant groups are suggested.

The paper will be of interest to a wide readership but the authors are asked to consider the following points before the manuscript can be accepted for publication:
The last sentence of the abstract states „widespread human dependence on these plants (plants containing psychoactive metabolites) for survival….“. I can not see why plants containing psychoactive metabolites are necessary for survival and the authors provide no arguments for this thesis. In my opinion this statement is too strong. There is no doubt that psychoactive compounds play an important role in human communities world wide but I would not go so far as to say that they are important for survival.

In the Results section lines 170 – 172 state that psyoactive compounds have evolved multiple times during angiosperm evolution. There is increasing evidence showing that plant metabolites are also biosynthesized by endophytic fungi. In some cases there is even mounting evidence for the hypothesis that endophytic fungi are the true source of these compounds such as for ergot alkaloids in the plant family Convolvulaceae. Nowhere do the authors comment on this fact. When analyzing the evolutionary origin of metabolites in the plant kingdom it is important to know whether plants are the real sources for these compounds. At least this point should be mentioned by the authors.

When covering sedative and anxiolytic compounds the reader would for sure like to hear something about Kava-Kava (Piper methysticum) that is so famous for its anxiolytic properties. There has been a long debate on the safety of phytopharmaca based on this plant and it is currently again being used in the EU. I would have expected a few lines on this important plant but it only appears in Table 1 without being mentioned anywhere in the paper.

I find it rather risky to state that „Europeans use plants with anxiolytic and antidepressant effects more than other ethnic groups due to the Western notion of happiness“ („Western assumption of individual happiness vs. happiness in other cultures arising more from social interdependence)“. This may be true for the modern world but use of psychoactive plnts has evolved over thousands of years. Has the European notion of individual happiness vs. happiness arising from social interdependence always been like this or has it evolved as an „individual happiness“ only in the last two or three centuries based on the philosophy of the era of enlightenment? If making statements on the notion of happiness in the European world one must look back into history and not judge from what we see today. I find some of these statements too simple and too superficial and hence dangerous.

Overall I find the paper interesting but it needs a revision according to the points raised above before it can be accepted for publication.

---

## Round 0.2 · Minor Revisions

Dear authors,

As you know I have published a few reviews in your field.

You mentioned my 2003 paper, but apparently you did not remember its main conclusion, which is relevant to your review. It is apparent that certain secondary metabolites occur in phylogenetically non related groups. This can be due to convergent evolution. We provided good evidence that in several instances the genes which underlie a pathway are commonly present but only expressed in certain groups.

Maybe you could consider the following publications, which are relevant to your review:

Van Wyk, B.-E., Wink, M. 2015: Phytomedicines, Herbal drugs and Poisons; Briza, -Cambridge University Press, Cambridge, USA
Wink, M. & Van Wyk, BE (2008): Mind-altering and poisonous plants of the world. Timber Press
Wink, M 2016 Evolution of Secondary Plant Metabolism. In: eLS. John Wiley & Sons, Ltd: Chichester.DOI: 10.1002/9780470015902.a0001922.pub3
Wink, M. 2016. Evolution, diversification, and function of secondary metabolites. In: Encyclopedia of Evolutionary Biology, 4, doi:10.1016/B978-0-12-800049-6.00263-8
Wink, M. (2013) Evolution of secondary metabolites in legumes (Fabaceae). South African Journal of Botany, 89, 164–175
Wink, M. F. Botschen, C. Gosmann, H. Schäfer and P. G. Waterman: Chemotaxonomy seen from a phylogenetic perspective and evolution of secondary metabolism. In Wink, M. (Ed.); Biochemistry of plant secondary metabolism, Blackwell, Annual Plant Reviews Vol. 40, 2nd ed., 2010
Wink, M. (2008) Plant secondary metabolism: Diversity, function and its evolution. Natural Products Com-munications 3, 1205-1216

Regards,
Michael Wink

---

## Round 0.3 · accepted · Accept

Dear authors

Thanks for the revision. Congratulation, your ms can now be accepted

Regards

Michael Wink
Academic editor